# Rules of hierarchical melt and coordinate bond to design crystallization in doped phase change materials

Jin Zhao [1,2,3], Wen-Xiong Song [1✉], Tianjiao Xin[1] & Zhitang Song [1✉]

While alloy design has practically shown an efficient strategy to mediate two seemingly conflicted performances of writing speed and data retention in phase-change memory, the detailed kinetic pathway of alloy-tuned crystallization is still unclear. Here, we propose hierarchical melt and coordinate bond strategies to solve them, where the former stabilizes a medium-range crystal-like region and the latter provides a rule to stabilize amorphous. The $Er_{0.52}Sb_2Te_3$ compound we designed achieves writing speed of 3.2 ns and ten-year data retention of 161 °C. We provide a direct atomic-level evidence that two neighbor Er atoms stabilize a medium-range crystal-like region, acting as a precursor to accelerate crystallization; meanwhile, the stabilized amorphous originates from the formation of coordinate bonds by sharing lone-pair electrons of chalcogenide atoms with the empty $5d$ orbitals of Er atoms. The two rules pave the way for the development of storage-class memory with comprehensive performance to achieve next technological node.

[1] State Key Laboratory of Functional Materials for Informatics, Shanghai Institute of Microsystem and Information, Chinese Academy of Sciences, Shanghai 200050, China. [2] School of Physical Science and Technology, Shanghai Tech University, Shanghai 201210, China. [3] University of Chinese Academy of Sciences, Beijing 100049, China. ✉email: songwx@mail.sim.ac.cn; ztsong@mail.sim.ac.cn

Phase-change random-access memory (PCRAM) is one of the most mature emerging nonvolatile memory technology, which is expected to achieve a processing-in-memory architecture applicable to big data[1], artificial intelligence[2,3], and other fields[4]. Being the focus of research and development for the chip industry and academia, PCRAM utilizes the ultrafast transition (~ns) between the high-resistance amorphous and low-resistance crystalline phases of phase-change materials (PCMs) to store information[5]. It has the advantages of nonvolatility, three-dimensional integratability, multibit memory, good scalability, and compatibility with CMOS process[6]. To fabricate commercial PCRAM, writing speed and data retention are two key performances, which are ~50 ns and 80 °C for commercially used $Ge_2Sb_2Te_5$ (GST), respectively[7]. But they are still currently fall short of the expectations from an ideal PCM. Alloy design or doping impurity is a practical strategy to improve them. It is always believed that accelerating the speed is incompatible with the improvement of data retention, seeing the review[8]. However, many doped systems of lattice replacement do not present such contradictory[6,9–11], albeit some systems with mismatched dopant meeting the experience[12,13]. Therefore, doping impurity with lattice replacement is an effective approach to solve the contradictory.

Nevertheless, direct atomic detail of how dopant with lattice replacement influences the kinetic pathway is still missing. Recently, Sc-Sb-Te (SST) material was reported to offer a record-breaking speed of 0.7 ns[9]. It is explained by the octahedral local motif around Sc dopant to reduce the stochasticity of nucleation, but of no more kinetic details. Moreover, although Y element has less octahedral-like local pattern in amorphous[9,14], Y-Sb-Te (YST) material still can achieve one order of magnitude faster speed than GST[11]. While we can enable the slow GST crystallization process via ab initio molecular dynamics (AIMD) simulation[15,16], not any indication of nucleation was observed in YST by a long-time AIMD simulation. The relative slower speed of YST than GST predicted by simulation is thus not in accordance with the relative experimental values. In other doped systems, their impressive faster chip speeds than GST are also hard to recur their crystallization trajectories by simulations[6,9–11]. Thus, a gap of the inconsistent results to predict the relative speed from the simulation and experiment should be bridged.

To fill the gap, we should understand the practical nucleation process at first. In liquids, cooperative movement (CM), as well as the similar concept of cooperatively rearranging region is a general feature as particle moves, which is manifested by both simulations[17–21] and experiments[22,23]. By monitoring the crystallization process, the CM process is necessary and monitored in a good metallic glass-former CuZr, where more crystal-like rejuvenated disorder states should be encountered before nucleation[24]. On account of the formation of strong bonds in the doped PCMs, the isolated dopants, even with octahedral motifs, make the CMs more difficult and impedes the critical nucleation of many crystal-like atoms connected[15]. Although reference has manifested that it is stable, compared with other regions, for the artificial embryo of a heterogeneous crystallite during a short-time melting simulation[25], it is unpractical to sow them using the sputtering technique and the embryo can melt as it undergoes enough time at high temperature. Here, we propose a hierarchical melt concept to produce a medium-range crystal-like region acting as a precursor, which is stabilized by several neighbor foreign atoms as less heat is provided, similar to the intention of low-voltage-incubation operation to form prestructural ordering[7,26].

On the other hand, why dopants stabilize amorphous still puzzles us, unknown of which challenges the choose of dopant to improve data retention. Generally, a criterion is used by choosing metals with high cohesive energy or high melting point[9]. While some elements, such as Al[27], Ga[28], In[29], and Sn[30], present low cohesive energy or low melting point, they, to our surprise, can stabilize the amorphous well. This gives us a hint that it probably has a deeper reason. Noteworthily, chalcogenide elements have electron-redundant nature or lone-pair electrons[31–33], which can be filled by the empty orbitals of neighbor atoms to form coordinate bond[32,33]. Thus, forming extra coordinate bonds to chalcogenide elements provides a way to stabilize amorphous.

In this work, we utilize hierarchical melt and coordinate bond concepts to design better comprehensive performance of PCMs. The hierarchical melt can be achieved by controlling the operation process of providing less heat (short pulse), whose schematic diagram is shown in Fig. 1a. The special electrical operation process of high-speed test is shown in Supplementary Fig. 1. In order to choose dopants with lattice replacement, we focus on metals, whose degree of mismatch with parental $Sb_2Te_3$ is summarized in Fig. 1b and the detailed mismatch values are shown in Supplementary Table 1. It shows that erbium (Er) has the least mismatch. It also has stable cubic ErTe and $Er_2Te_3$ phases in phase diagram. Moreover, the empty $5d$ orbitals of Er can be filled by lone-pair electrons of Te atoms, manifested by partial density of states (pDOS) and crystal orbital Hamilton populations (COHP) shown in Fig. 1c–f. In the following, the excellent comprehensive performance of Er-doped $Sb_2Te_3$ is exhibited and explained.

## Results

**Device performance.** Our investigation starts by experimentally testing the device performance of Er-doped $Sb_2Te_3$, as shown in Fig. 2. To evaluate data retention, we utilize the resistance-temperature or -time (R-T or R-t) curves, as shown in Fig. 2a, b. The sharp drop of R-T curves is defined as crystallization temperature ($T_c$), and the tested $T_c$ values of four compositions, $Er_xSb_2Te_3$ ($x = 0.31, 0.41, 0.52, 0.76$), are 203, 235, 256, and 289 °C, respectively. It demonstrates that Er can improve amorphous stability significantly. It is noted that the resistance is increased for both amorphous and crystalline phases after doping Er, which is helpful to boost heat efficient by providing lower RESET current[34]. Moreover, the 10-year (or 100-year) data retention is estimated according to the Arrhenius equation, $t = \tau \exp(E_a/k_B T)$, as shown in Fig. 2b, which are 103 °C (91 °C), 129 °C (117 °C), 161 °C (151 °C), and 198 °C (189 °C), respectively. They are much higher than the commonly used PCMs, such as GST(~80 °C)[35], SST (~85 °C)[9], even nonmetal-doped $C - GST$[6] and $N - GST$[36], which meets temperature requirement of most nonvolatile applications[28].

By comparing the device performance of four components and pure $Sb_2Te_3$ (Supplementary Fig. 2), $Er_{0.52}Sb_2Te_3$ (EST) has better comprehensive performance and is chosen for the following electrical tests using a T-shape cell, whose cell schematic diagram is shown in Supplementary Fig. 3. Fig. 2c shows the SET-RESET windows using the resistance–voltage (R-V) curves, whose high/low-resistance ratio ($R_{RESET}/R_{SET}$) is about two orders of magnitude and meet the requirement of ON/OFF ratio used in PCRAM. It is noted that EST has an operating speed of 3.2 ns, which is faster than most PCMs reported[10,11], albeit still slower than SST. Its SET ($V_{SET}$) and RESET ($V_{RESET}$) voltages are 3.2 V and 4.4 V, respectively, which are lower than GST[9] and is responsible for its low-power consumption. In Fig. 2d, the tested power consumption of $Er_{0.52}Sb_2Te_3$ is 1.14 nJ using the 500-ns-width current pulse, which is lower than the GST[37] (9.28 nJ), Ti-$Sb_2Te_3$[10] (TST, 3.12 nJ), and SST[9] (1.68 nJ). The physical reason may originate from the grain refinement, discussed in Supplementary Figs. 4–5. Fig. 2e shows the endurance of ~$10^7$ cycles by

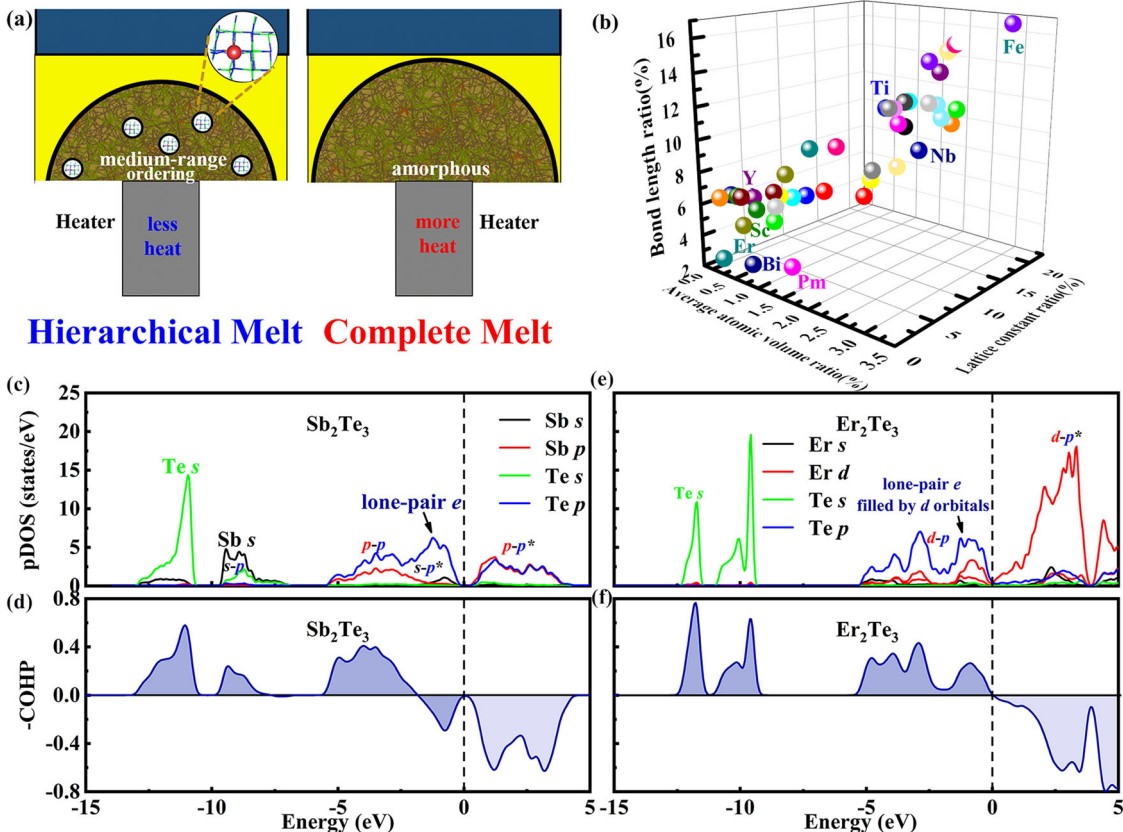

**Fig. 1 Alloy design. a** The schematic diagrams of device cells with different amorphous states as different power is provided. The left part has many medium-range ordering regions as less heat is provided, while the right part is a fully disordered amorphous state as more heat is provided. **b**, The lattice mismatch between metal tellurides and $Sb_2Te_3$, where Er has the least mismatch. **c–f** Evidences of orbital interactions by calculating the pDOSs and COHP of $Sb_2Te_3$ and $Er_2Te_3$. **c, e** The pDOSs for various orbitals. **d, f** The -COHP curves for $Sb_2Te_3$, and $Er_2Te_3$, respectively.

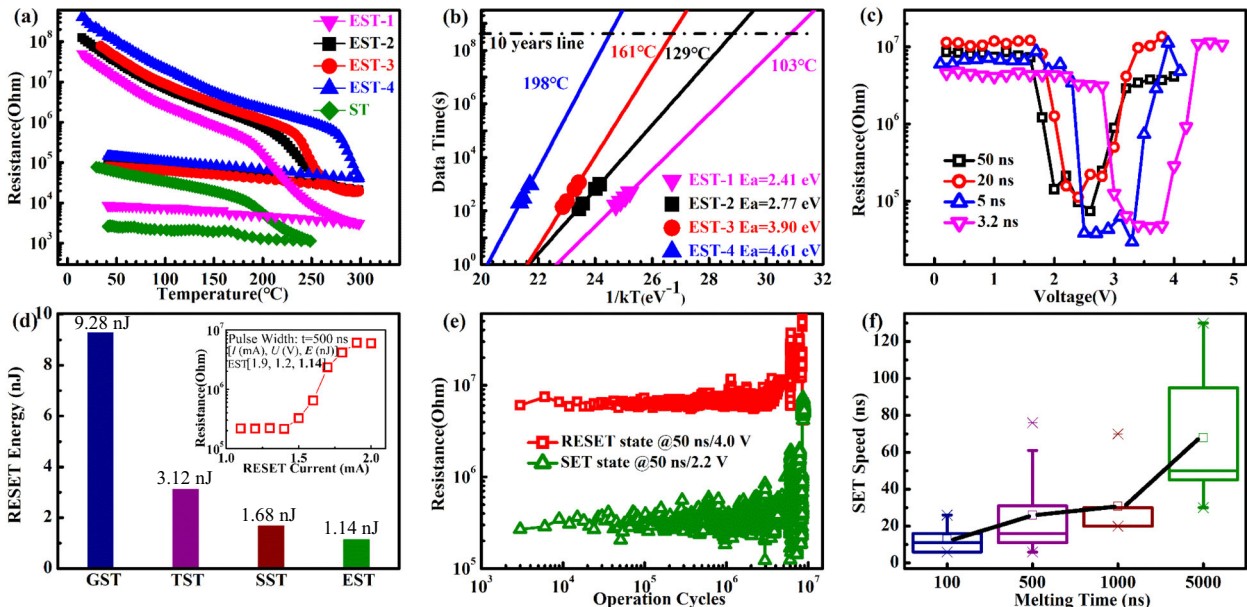

**Fig. 2 Device performances. a** Temperature dependence of the resistance for $Sb_2Te_3$ and $Er_xSb_2Te_3$ films at the same heating rate of 10 K min$^{-1}$. **b** The 10-year (or 100-year) data retention temperature and activation energy of crystallization are deduced from the extrapolated fitted lines based on the failure time versus reciprocal temperature. **c** The switching property characterized by the SET-RESET windows for $Er_{0.52}Sb_2Te_3$. **d** Using current pulse, the tested power consumption of $Er_{0.52}Sb_2Te_3$ is compared with GST[37], TST[10], and SST[9]. **e** Operation cycles of $Er_{0.52}Sb_2Te_3$. **f** Bbox and whisker graph of SET speed under different melting time, where 80 samples are used. All box and whisker plots represent the median (central line), 25th–75th percentile (bounds of the box) and 5th–95th percentile (whiskers).

imposing two appropriate SET and RESET voltage pulses alternately, which is one order of magnitude higher than GST[38]. The endurance may be improved further using the confined structure, manifested in GST with over $10^{12}$ cycles[39]. Moreover, the endurance using a shorter pulse (SET: 25 ns/2.5 V, RESET: 15 ns/3.8 V) achieves $2 \times 10^5$ cycles, as shown in Supplementary Fig. 6. Thus, our designed EST has great potential for the storage-class memory applications.

In order to manifest the necessary of CM process during the crystallization, we take a statistic of SET speed using the different melting time, whose operation details are shown in Supplementary Fig. 7. Fig 2f presents the box chart of speed. It exhibits the result that the longer melting time, the slower SET speed. It is because the longer melting time makes the amorphous more disorder. The system needs more CM processes, i.e., more time, to complete critical nucleation. It gives us a hint that less disorder amorphous or pre-existing medium-range crystal-like cluster can shorten the crystallization time. In the following, we focus on (i) manifesting the cationic positions replaced by Er atoms from the experiment, and (ii) uncovering the calculation or simulation details how dopant stabilizes the amorphous and accelerates the crystallization.

**Direct atomic evidence of Er replacing cationic positions**. From the previous mismatch analysis, it predicts that the least mismatch Er prefers replacing lattice position, due to the least strain energy, which can avoid the separation of dopant atoms. From the calculations, Er locating at the cationic position has the substitution energy of −1.97 eV and is ~0.5 eV lower than in the anionic position. The details are shown in Supplementary Fig. 8. Fig. 3 provides the experimental evidences by showing the atomic arrangement and elemental distribution in the crystalline phase of EST using in situ transmission electron microscope (TEM)

measurement in a [110] oriented plane at 280 °C. Fig. 3a shows the dark field STEM image. A vacancy ordering layer (VOL) is found, but not sharing two Te-terminated boundaries for forming a vdW gap. The intensity profile of the marked rectangle region in Fig. 3a can be seen below. The distance of two Te planes is about ~4.4 Å, which is consistent with the reported value of cubic phase[40], 4.1~4.5 Å. It illustrates that EST is still a cubic phase, whose structure difference compared with hexagonal phase is shown in Supplementary Fig. 9. It should be mentioned that the cation intensity near the VOL, marked by star, has similar value with its neighbor anion position, which indicates that Er with large atomic number has high content in this position.

Fig. 3b-d identify the distribution of three elements via the atom-resolved energy-dispersive X-ray spectroscopy (EDX). We clearly resolve the distribution of Er dopant in the [211] direction, but the blurry resolution in the [110] direction, as shown in Fig. 3b, which may attribute to the complicated shape of Er $d$ orbitals. Fig. 3c shows the distribution of Sb atoms, whose resolution is clear in the Er-poor region. In order to verify both Er and Sb atoms in the Er-enrich region, we check the intensity of Er and Sb along the cyan line 2 in Fig. 3a, as shown in Fig. 3f. The coupled intensity peaks of Er and Sb demonstrates that they are both in the cationic position. In the XPS test, the redshift of the binding energy of Te $3d$ orbital further manifests the interaction between the Er atoms and anionic Te atoms, as shown in Supplementary Fig. 10. Therefore, we can make a conclusion that Er locates at cationic positions.

**Stabilizing anionic Te atoms by forming coordinate bonds**. Subsequently, we uncover the reason why dopant stabilizes the amorphous via chemical bonding analysis. Fig. 4a-b show the pDOS and COHP in the amorphous. Similar to the scenario in the crystalline $Sb_2Te_3$ phase shown in Fig. 1, all bonding $p_{Sb} - p_{Te}$

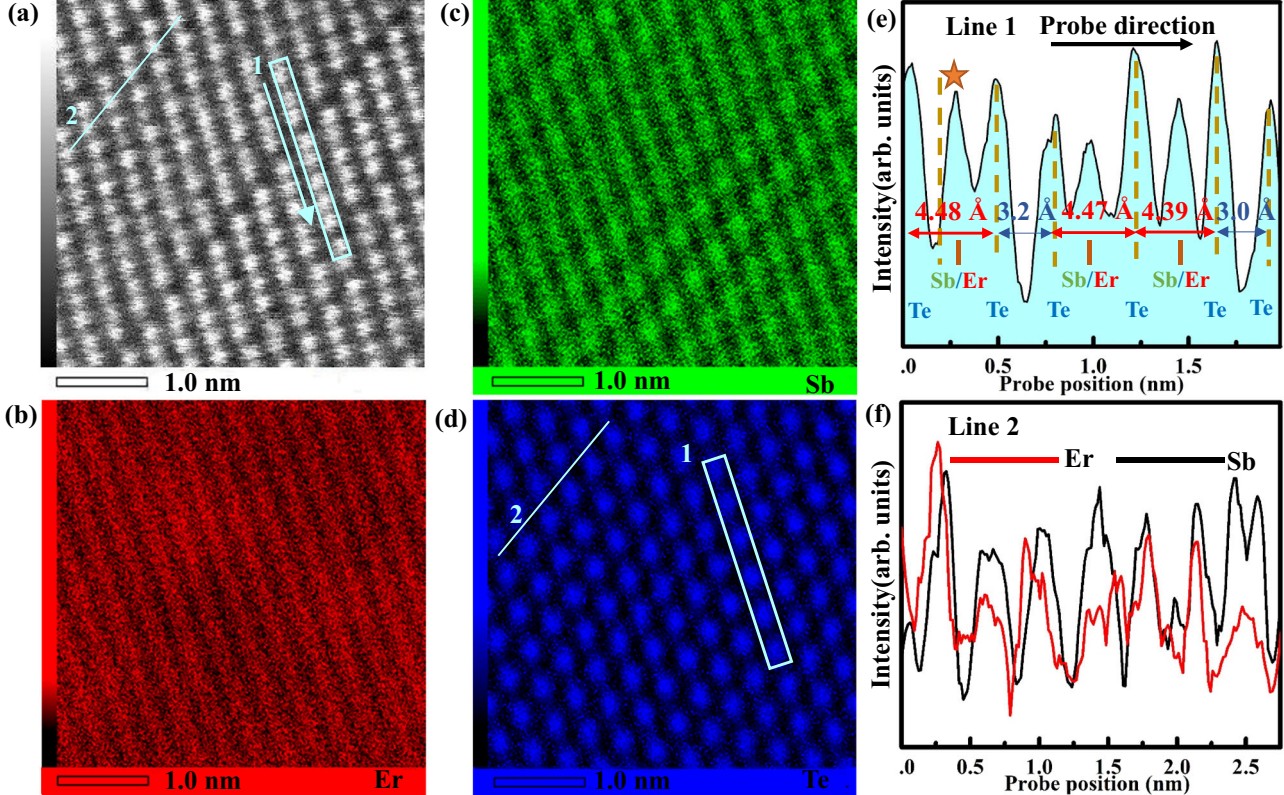

**Fig. 3 Atomic structure of cubic EST phase at 280 °C. a** Dark field image. **b–d** The atomic resolution map images of Er (red), Sb (green) and Te (blue) atoms taken from (**a**). **e** The intensity profile of marked rectangle region in line 1. **f** The contrast intensity of Er and Sb along line 2.

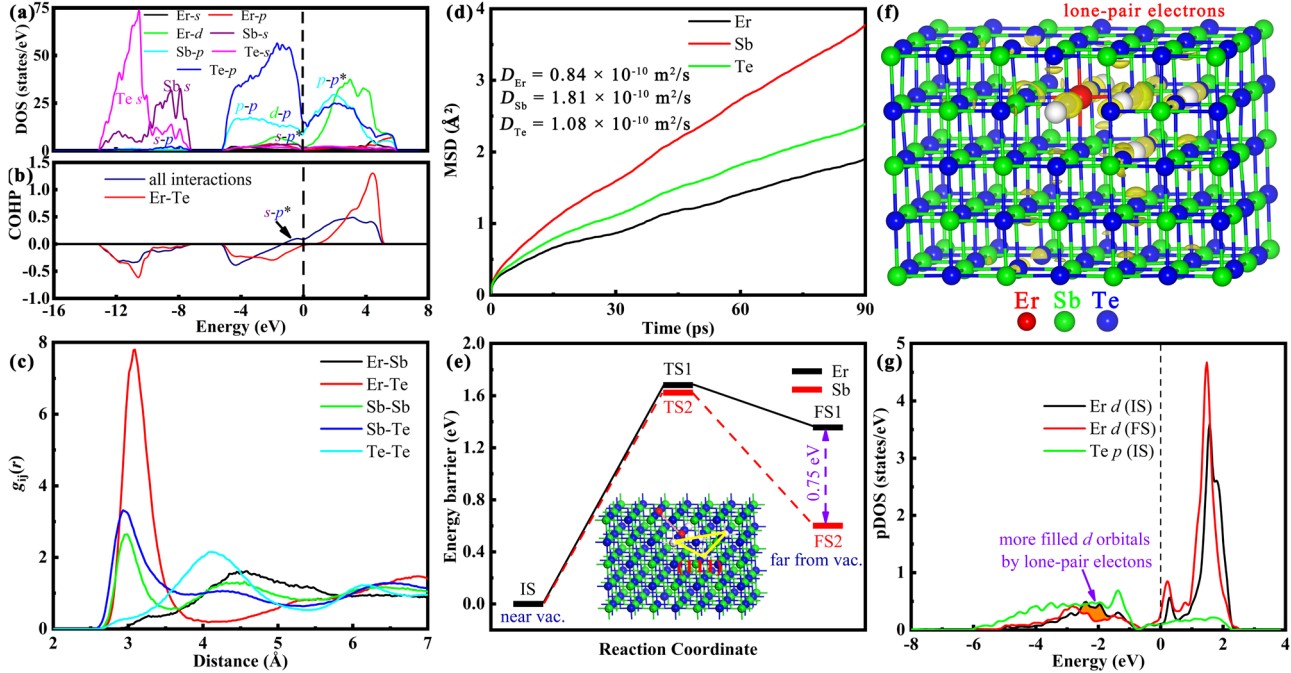

**Fig. 4 Forming coorindate bonds by filling the empty 5d orbitals of Er atoms with the lone-pair electrons of Te atoms.** It stabilizes both amorphous (**a–d**) and crystalline phases (**e–g**). **a** The pDOS of Er, Sb, and Te atoms with various orbitals. **b** The COHP of average values of all interactions and Er−Te interaction, respectively. **c** The PCFs of Er−Sb (black), Er−Te (red), Sb−Sb (green), Sb−Te (blue), and Te−Te (cyan) pairs. **d** The MSDs for Er, Sb, and Te atoms. **e** The energetic profile of diffusion barrier for Er (black) or Sb (red and dash), where a model with a four-vacancy-aggregated cluster in the (111) plane is shown in the inner graph. **f** The partial charge in the DOS region of $[-3, 0]$ eV in the IS structure, where 3D isosurface value is set as 0.2 e$\text{Å}^{-3}$. **g** The partial DOS of Er $d$ (black) and Te $p$ (green) orbitals in the IS structure, while the Er $d$ (red) orbital in the FS structure also shown for comparison. It is noted that the pDOS of Te $p$ orbitals is divided by Te number (48 herein).

orbitals below Fermi level and all antibonding $p_{Sb} - p_{Te}*$ orbitals above Fermi level. The $s$ orbitals of both Sb and Te atoms mainly locate at deep energy levels, albeit a weak hybridization between the $s$ orbitals (Sb) with $p$ orbitals (Te), where the antibonding $s - p*$ orbitals is just below Fermi level. Remarkably, we find the lone-pair electrons of Te atoms fill the empty $d$ orbitals of Er atoms, the same as the above discussion in the ErTe material, which stabilizes the amorphous of anionic Te atoms.

Fig. 4c shows the pair correlation functions (PCFs) of Er−Sb, Er−Te, Sb−Sb, Sb−Te, and Te−Te pairs in the amorphous. For the first peak, we find that the Er−Te is the highest. To our surprise, Te−Te homopolar bonds are almost disappeared according to its weak first peak, against the fact of huge homopolar bonds in the common amorphous PCMs[41]. The disappeared Te−Te homopolar bonds may attribute to the preferred Er−Te interaction. The observed Sb−Sb homopolar bonds are because of weak Er−Sb interaction with lower first peak. Moreover, the dynamic property, such as diffusion coefficient, can further verify the stabilized Te atoms. Fig. 4d shows the result of mean-square displacement (MSD) for Er (black), Sb (red), and Te (green) atoms. It shows that Er has the least displacement, because of its maximum coordinate number shown in Supplementary Fig. 11 and many coordinate bonds formed with Te atoms. The displacement of stabilized Te atoms is about half of Sb atoms, compared with their similar values in the GST[41,42]. Therefore, a conclusion is made that the formed coordinate bond is the essential reason of the stabilized amorphous.

On the other hand, the formed coordinate bond also exists in the crystalline phase and has a significant effect on PCM properties. It has been proved that Te atoms near vacancy traps electrons near Fermi level[43], as well as lone-pair electrons near Fermi level shown in Fig. 1c-e. Thus, we can predict that Er

prefers the cationic position near vacancy by sharing the empty $5d$ orbitals with the neighbor lone-pair electrons of Te atoms to form coordinate bonds. It is in line with the above experiment result of high content Er near the VOL. The VOL (or Te-terminated boundary) is also stabilized at the same time. It can refine grains seriously, because our recent work has proved that the stable Te-terminated boundary is the reason of small grain size in some PCMs[44]. The increased number of grain boundary will hinder the charge transport property[44–46] and scatters phonons[38], which boost heat efficiency and more evidences see Supplementary Figs. 4, 5.

Next, we provide the calculation evidences of the formation of coordinate bond and the stabilized Te-terminated boundary. Fig. 4e shows the diffusion barrier of Er (black) or Sb (red) migrating from near the four aggregated vacancies in the (111) plane to the inner vacancy. Remarkably, the final state (FS) of Er has the energy of ~1.4 eV higher than the initial state (IS). It is still 0.75 eV higher than that the FS of Sb migration, both of which have similar migration barrier. The much unstable FS state illustrates that VOL traps Er movement seriously. To unravel the further reason, we compare the pDOS difference between the IS and FS states, as shown in Fig. 4g. We find that more lone-pair electrons fill the $5d$ orbitals of Er atoms in the IS state, because lone-pair electrons prefer Te atoms near vacancies, manifested by the partial charge distribution in the region of $[-3, 0]$ eV, as shown in Fig. 4f.

**Hierarchical melt tuned crystallization.** Finally, to uncover how Er dopant influences the crystallization kinetics, we monitor the process without any embedded seeds, where a low concentrate model, $Er_4Sb_{72}Te_{108}$, is used. Firstly, we carry out a simulation of a complete melt model using the common melt-quench method. The sample melts fully at 3000 K for 120 ps, and then quenches to

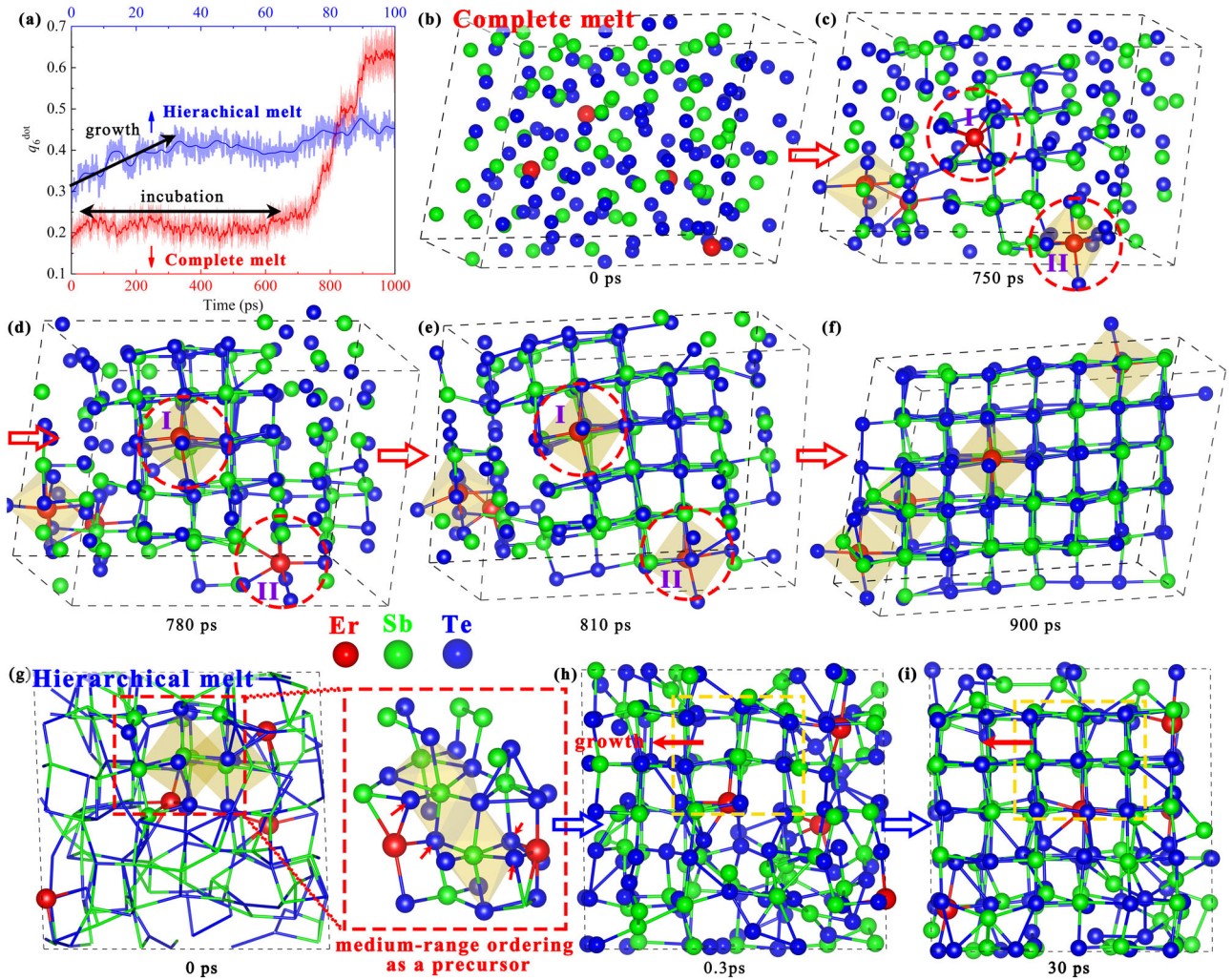

**Fig. 5 Hierarchical melt tuned crystallization. a** The evolution of $q_6^{dot}$ value during the crystallization for the complete melt (red) and hierarchical melt (blue) models. In the complete model, **b–f** show the snapshots at 0 ps, 750 ps, 780 ps, 810 ps, and 900 ps, respectively, where the octahedral environment around Er atoms is characterized by orange octahedral cages. The crystal-like atoms and their neighbor atoms are connected by bonds with the cutoff of 3.6 Å. The red circles in **c–e** emphasize the alterable Er local environments. In the hierarchical melt model, **g–i** show the snapshots at 0 ps, 0.3 ps, and 30 ps, respectively.

600 K for crystallization annealing. Fig. 5a shows the structural evolution (red) of EST characterized by $q_6^{dot}$ parameter during the crystallizaiton, which monitors the ordering of local environment, defined in the method. Fig. 5b-f are five snapshots, where we only connect the bonds among crystal-like atoms defined as $q_6^{dot} > 0.45$, as well as its neighbor atoms.

At the beginning, as shown in Fig. 5b, four Er atoms are randomly distributed in the system. At 750 ps, a $Sb_7Te_{11}$ nucleus is shown in Fig. 5c. Near the nucleus, a Er atom without octahedral pattern, marked by $Er_I$, is monitored. As the nucleus grows, the local environment of $Er_I$ atom changes to octahedron and becomes one part of the small crystallite, as shown in Fig. 5d. It is noted that a Er atom with octahedral environment, marked by $Er_{II}$, is found far from the nucleus at 750 ps, while the $Er_{II}$ motif becomes nonoctahedron at 780 ps as the nucleus grows up to nearby it. Interestingly, the local pattern of $Er_{II}$ finally changes to octahedron again at 810 ps and it integrates into the crystallite, as shown in Fig. 5e. Therefore, the alterable local motifs of Er atoms manifest the necessary of a series of CMs during the crystallization. Finally, all Er dopants locat at the cationic positions, as shown in Fig. 5f, agreeing with above theoretical prediction and experimental observation.

However, the incubation time in the complete melt model is much longer than the reference simulation result of GST[16,42]. It conflicts with our chip test that EST has much faster speed than GST. In the complete melt model, all atoms are fully disordered. The stabilized Te atoms by Er slow down the CMs, as well as the formation of critical nucleation. It probably exists another more practical pathway to achieve the faster speed of EST from the simulation. It is noted that, in the actual high-speed test, it is hard to obtain the high speed by directly annealing the as-deposited amorphous state (complete model). In fact, a much shorter pulse (or less heat) should be provided to melt the crystalline state for the high-speed chip test, which achieves a hierarchical melt discussed at the very beginning.

In the following, we utilize the hierarchical melt model to correct this mistake of relative speed prediction between the simulation and experiment. Based on the fact of Er preferring the canionic positions near VOL revealed above, we construct a crystalline model that two Er atoms are close to VOLs, as shown in Supplementary Fig. 12. The model is melt at 1500 K. At 1.5 ps, we find a medium-range region that is crystal-like and has two regular Sb-center octahedrons, which is stabilized by the two Er atoms near the VOLs, as shown in Fig. 5g. The enlarged local

structure shows that Er atoms stabilize four Te atoms constituting two octahedrons. Annealing at 600 K, this medium-range region gradually grows up without obvious incubation, which is different from the complete model. At 30 ps, the nucleus grows to a big one, ableit some region still disorder because of the kinetic constraint around the relative unstable Er atoms that slows down CMs. Therefore, the hierarchical melt model provides a right simulation prediction that EST has faster speed than GST.

## Discussion

The hierarchical melt and coordinate bond concepts are general to understand how replacement dopants influence the amorphous stability and crystallization processes. The above calculation evidence has shown that the bonds of chalcogenide atoms become stronger by sharing its lone-pair electrons with the empty $5d$ orbitals of Er atoms. It is also suited to other dopant with empty orbitals, such as transition metals with empty $d$ orbitals, to stabilize amorphous PCMs[9–11]. It is the same scenario in Al[27], Ga[28], In[29], and Sn[30] with more empty $p$ orbitals to stabilize the chalcogenide glass, albeit their lower cohesive energy or low melting point compared with the transition metals.

In addition, similar to Er, a little mismatch metals, such as Sc, Y, and Ti that have the impressive transition speed reported, will prefer the location near VOL and present hierarchical melt as a little heat is provided. To verify this prediction, we utilize the model used in Fig. 4e to calculate the diffusion barrier of Sc, Y, and Ti dopants. We obtain the similar results to EST that these dopants locating at positions far from vacancies have much higher energy, as shown in Supplementary Fig. 13, because of less lone-pair electrons filling their empty $d$ orbitals. Further, medium-range crystal-like regions stabilized by these dopants will act as precursors to accelerate the crystallization.

It has been reported the emphasized octahedral motifs around dopants[3,9,14]. However, it ignores the necessary CM processes before the formation of critical nucleus. On the contrary, the much stable isolate octahedral motif still impedes the local structure movement and slows down the formation of critical nucleus. It is the essential reason why the wrong relative speed is predicted in the Er-, as well as Sc-, Y-, and Ti-doped systems using the complete melt models, albeit stable octahedral motifs observed in these systems.

Finally, we should discuss the effect of Er $f$ orbitals on the calculation results, because the above calculations treat Er $4f$ electrons as core states. In fact, current density functionals can not handle $f$ electrons well, due to self-interaction errors. A routine way to describe the localized $4f$ electrons by placing them in the core. As a contrast, we consider the $4f$ electrons as valence electrons to calculate the relative energy and partial DOSs of the same IS and FS1 structures in Fig. 4e, as shown in Supplementary Fig. 14. Without Hubbard $U$ added, the relative energy difference of FS1 and FS2 is 0.44 eV, which is about half of the 0.75 eV in Fig. 4e. Using Hubbard $U$ (4 eV) to correct $4f$ electrons localization, the relative energy of 0.82 eV is close to the value in Fig. 4e. Both scenarios prove the unstable FS1 structure filled less lone-pair electrons in the Er $5d$ orbitals than the stable IS structure. Many other calculations aslo obtained reasonable calculation results as Er $4f$ electrons in the core[47–50]. It demonstrates that the pseudopotential performs well as the Er $4f$ electrons included in the core.

In summary, based on the proposed rules of hierarchical melt and coordinate bond, we design the $Er_{0.52}Sb_2Te_3$ material, whose excellent device performance surpass most PCMs have been reported. It has the potential for future application in storage-class memory: 3.2 ns operation speed, 161 °C data retention, ~$10^7$ endurance, 1.14 nJ power consumption, and 0.41% density-change

rate. The two rules solve the contradiction of writing speed with data retention, and provide a general way to design storage-class memory with comprehensive performance.

## Methods

**Characterizing film samples**. Radio-frequency magnetron co-sputtering method with Er and $Sb_2Te_3$ targets is used to deposit the $Sb_2Te_3$ films and Er-doped $Sb_2Te_3$ films. The components of the designed films were identified by sputtering power and measured by energy-dispersive spectroscopy. Films with a thickness of 200 nm were deposited on $SiO_2/Si(100)$ substrates for resistance-temperature and X-ray diffraction (XRD) tests. The resistance by a function of the temperature was performed in a vacuum chamber with the heating rate of 10 °C min$^{-1}$, and isothermal change in resistance with elevated temperature was recorded to estimate the 10-year data retention. The X-ray reflectivity experiment (Burker D8 Discover) was employed to test the density change before and after the crystallization of ~40 nm thickness films. XRD was adopted to characterize the lattice information of films. X-ray photoelectron spectroscopy experiment was used to evaluate the bonding situation. Then the microstructure of these samples was studied by JEM-ARM 300 F Transmission Electron Microscope and High-resolution transmission electron microscope, as well as in selected area diffraction mode.

**Fabrication PCRAM devices**. T-shaped phase-change random-access memory devices with tungsten plug bottom electrode contact (BEC, diameter = 190 nm) are fabricated using the 0.13 μm node complementary metal-oxide semiconductor technology. The 60 nm-thick phase-change material and 10 nm-thick TiN as an adhesion layer are deposited using the sputtering method over the tungsten electrode, then 300;nm-thick Al is deposited by using UMS500P Electron Beam Evaporation to form top electrode. The current−voltage (I − V), resistance−voltage (R − V), and endurance tests are carried out using a Tektronix AWG-4012 and 5002B arbitrary waveform generator and a Keithley-2400 meter parameter analyzer. The cell resistance after applying voltage pulses was recorded at a constant read voltage of 0.1 V.

**First-principle calculations**. First-principle calculations are carried out using VASP package[51]. The Kohn–Sham equations are solved using the projector augmented wave method[52] and Perdew−Burke−Ernzerhof with van der Waals correction (PBE-D3)[53,54] generalized gradient approximation functional[55] with the kinetic energy cutoff of 388 eV. The valence electrons are $5p^65d^16s^2$ for Er_3, $5s^25p^3$ for Sb, and $5s^25p^4$ for Te in the main text. Molecular dynamics is carried out to study the effect of dopant on crystallization kinetics at 600 K, where we use a time step of 3 fs with Parrinello-Rahman barostat and Langevin thermostat.

**Order Parameter**. The degree of crystallinity is judged by $q_6^{dot}$ parameter, defined in the following[56],

$$Q_l = \left[ \frac{4\pi}{2l+1} \sum_{m=-l}^{l} \left| \sum_{b=1}^{N_b} Y_{lm}(\theta_b, \phi_b) \right|^2 \right]^{\frac{1}{2}} / N_b \tag{1}$$

where the spherical harmonics $Y_{lm}$ describes the local order of the centered atom surrounded by its nearest-neighbor atoms. The summation in Eq. (1) runs over all $N_b$ bonds in the first shell within a cutoff of 3.6 Å in this work.

However, Steinhardt's order parameter is not convenient for the condition of multiple crystallites instead of a single crystalline nucleus. A local version of $q_l$ can be defined for each atom in the following vector:

$$\boldsymbol{q_l}(i) = \begin{pmatrix} q_{l,l} \\ q_{l,l-1} \\ \dots \\ q_{l,-l+1} \\ q_{l,-l} \end{pmatrix} = (q_{lm}(i))_{m=-l,l} \tag{2}$$

$$q_{lm}(i) = \frac{1}{N_i} \sum_{j \in \Omega_i} f_{ij} Y_{lm}(ij) \tag{3}$$

A radial cutoff function $f_{ij}$ is introduced to smooth the boundary:

$$f_{ij}(r) = \begin{cases} 1 & : r \leq r_1 \\ \frac{1}{2}\left\{ \cos\left[ \frac{\pi(r-r_1)}{(r_2-r_1)} \right] + 1 \right\} & : r_1 < r \leq r_2 \\ 0 & : r > r_2 \end{cases} \tag{4}$$

The exponents in $f_{ij}$ were set to $r_1 = 3.2$ Å and $r_2 = 3.6$ Å. The norm of $\boldsymbol{q_l}(i)$ is a local $Q_l$ version for an atom:

$$q_l(i) = \sqrt{\frac{4\pi}{2l+1}} \| \boldsymbol{q_l}(i) \| \tag{5}$$

The order parameter $q_l^{dot}$ is defined based on the bond order correlation $C_{ij}$ between neighboring atoms, first introduced by Frenkel and coworkers[57].

$$C_{ij} = \frac{\boldsymbol{q}_l(i) \cdot \boldsymbol{q}_l^*(j)}{\| \boldsymbol{q}_l(i) \| \cdot \| \boldsymbol{q}_l^*(j) \|} \tag{6}$$

The order parameter $q_l^{dot}$ is the averaged sum of the bond order correlation $C_{ij}$, which is defined as the dot product of $q_l(i)$ and the complex conjugate of $q_l(j)$, divided by the rotationally invariant norm of the two vectors:

$$q_l^{dot}(i) = \frac{1}{N_i} \sum_{j \in \Omega_i} f_{ij} C_{ij} \tag{7}$$

The radial cutoff function $f_{ij}$ has the same form as above with the cutoff radius.

**COHP analysis**. High-precision calculations are carried out using VASP package, and then implement the pCOHP bonding analyses using LOBSTER setup[58].

**Energy barrier calculation**. The migration barrier is calculated using the Stochastic Surface Walking (SSW) method[59,60] via smooth surface walking along softened random directions. The explicit transition states of the pathways are located by the variable-cell double-ended surface walking (VC-DESW) method[61].

## Data availability

All data needed to evaluate the conclusions in the paper are present in the paper and/or the Supplementary Materials. The source data underlying Fig. 2 and Fig. 3 are provided as a Source Data file (DOI: 10.24435/materialscloud:cs-2a). Additional data related to this paper can be requested from the authors.

## Code availability

The first-principle calculation and simulations were carried out using VASP package. The migration barrier was obtained using the Stochastic Surface Walking method using the LASP package, whose website is at http://www.lasphub.com. The COHP analysis was performed by LOBSTER setup.

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

## Acknowledgements

Supported by the National Key Research and Development Program of China (2017YFB0701703, 2017YFA0206101, 2017YFA0206104, 2018YFB0407500, SQ2017YFGX020134), "Strategic Priority Research Program" of the Chinese Academy of Sciences (XDB44010200), National Natural Science Foundation of China (61904189, 62174168, 61874129, 61874178), Science and Technology Council of Shanghai (17DZ2291300, 18DZ2272800), Shanghai Pujiang Program (21PJ1415300).

## Author contributions

J.Z. performed the experiments and W.-X.S. carried out the calculations and simulations. T.X. took the TEM analysis. J.Z. and W.-X.S. wrote the paper. All authors discussed the results and commented on the manuscript. The project was initiated and conceptualized by W.-X.S. and Z.S.

## Competing interests

The authors declare no competing interests.
