## [Peer Review File · Nature Communications]

Designing crystallization to tune performances of phase-change memory: rules of hierarchical melt and coordinate bondREVIEWER COMMENTS

Reviewer #1 (Remarks to the Author):

J. Zhao et al. proposed a hierarchical melt and coordinate bond strategies to study the kinetic pathway of alloy-tuned crystallization in phase change material. They experimentally demonstrated Er-Sb₂Te₃ based phase change memories with 3.2 ns writing speed, ten-year data retention at 161 oC, 10⁵ endurance, 1.29 pJ power consumption and 0.41% density change rate. Some of the performances are among the best results in NVM field, like 3.2 ns operating speed and 1.29 pJ working energy. But some performances show obvious shortcoming, like 10⁵ endurance. The authors also conducted ab-initio calculation and discussed the crystallization kinetic of Er-Sb₂Te₃ material in-depth. In general, each part of this paper looks good, but the link between sections is weak. Here, I list one major issue and several technique issues for your reference:

Major issue:

In this work, the authors raised several critical issues in the field of phase change memory, including conflicted performances between writing speed and data retention, the detailed kinetic pathway of alloy-tuned crystallization, the gap between ab-initio simulation and experiment result, dopant selection criteria to improve data retention and speed etc. And they also provide plenty of experimental and simulation results to detailly discuss Er-Sb₂Te₃ material from multiple perspectives. However, the coupling between the critical issues, experimental results and ab-initio simulations is weak. In other words, the questions raised by authors are not fully addressed by the experimental and simulation results. For example:

(1) In experiment, the authors demonstrate 3.2 ns writing speed, but in simulation, the material is fully crystallized in 900 ps. The gap between experiment and ab-initio simulation still exists.

(2) The in-situ HRTEM shows very good atomic level resolution images of Er-Sb₂Te₃, but it only provides limited information that “Er locates at cationic positions”. It fails to provide a “direct atomic-level evidence” for “the detailed kinetic pathway of alloy-tuned crystallization”.

(3) The hierarchical melt and coordinate bond strategies is a good concept. But this concept should commonly exist in transition metal doped Sb₂Te₃ phase change materials, like TiSbTe, ScSbTe, YSbTe etc.

Therefore, in my opinion, the authors should enhance and rearrange their experimental results and simulation results to make them better support each other.

Technique issues:

1. The authors claimed that their ErSbTe-based phase change memory can serve as storage-class memory (SCM). As indicated by IBM research, there are 15 criteria to identify a SCM. The sub-10 ns

speed, pJ level power consumption and 10-year@161oC thermal stability well meet the corresponding requirements of SCM. However, the write endurance of ErSbTe-based memory is only 10⁵, which can hardly meet the lifetime requirement of SCM (10⁹ – 10¹² write/erase cycles). Therefore, I think the authors should identify a new application scenario for this device.

2. For the power consumption calculation, the authors used a steady-state calculation method by using $Q_t = (V_{reset}^2/R_{set}) \times t_{reset} + (V_{set}^2/R_{reset}) \times t_{set}$. This method is based on the constant resistances of ON and OFF state. However, this calculation is questionable, because the resistance of phase change memory dynamically changes during resistive switching. I highly recommend to use transient calculation method by capturing current waveform during voltage pulse duration, which is more convincing for operating energy evaluation. Or the authors can also use the calculation method by using a current pulse source as in their previous works (Science 366, 210–215 (2019). Nature Communications 5 (1), 1-6 (2014).).

3. In Fig. 1b, the authors mentioned that Er-Te has least mismatch with Sb₂Te₃ based on volume change rate calculation. I think their discussions in this part are weak. Because (1) lattice matching between two materials does not only depend on volume change rate, but also relate to other factors, like lattice constant, bond length, bond angle etc. (2) In Fig. 1b, the authors only included some transition metal elements, they missed some elements like Nb etc. for lanthanide series metals, they only calculated Er. So, it is hard to say Er-Te has the “least mismatch” with Sb₂Te₃.

The authors should provide solid calculation results to support their discussion. This is important, because Er is not commonly used in standard semiconductor process. It should provide obvious advantages to motivate us to introduce new elements in standard process.

4. In Figure 5, it can be seen that the Er atoms tends to form octahedrons. And two Er centered octahedrons stabilized medium-range region, which accelerates the crystallization. But in other doped Sb₂Te₃ phase change materials. But metal centered octahedrons are commonly formed in other phase change materials, like Ti-SbTe, Sc-SbTe, Y-SbTe etc. So, the authors should indicate the difference between Er-SbTe and other transition metal doped Sb₂Te₃ phase change materials.

Reviewer #2 (Remarks to the Author):

The manuscript by Zhao et al reports an alloy design of phase-change materials based on the idea on the hierarchical melt and coordinate bond. Even though there are some interesting results, the current version of manuscript is not suitable for publication due to the following reasons.

1. The authors mentioned that Er has the empty d orbitals. Does this mean 5d orbitals? On the other hand, Er is a lanthanide with partially filled f orbitals. In general, DFT calculations for f-electron materials are not straightforward. According to the Methods, the authors used just a conventional GGA function, so I wonder the reliability of their simulation results without using the DFT+U term.

2. Device performance of four different Er-Sb₂Te₃ compositions were compared in Fig.2(e) and Fig. S2. I would recommend adding the results for the pure-Sb₂Te₃ to show the effects of Er-doping.

3. In Fig.2(f), the endurance data is shown, and the authors mentioned that “the tested endurance over 10⁵ cycles, showing great potential for the storage-class memory applications”. I doubt that 10⁵ is sufficient to use as the storage class memory, but at least 10⁷ cycles are required.

Moreover, the authors applied 100ns/3.8V Reset and 300ns/1.8V Set pulses to switch the phases for the endurance test. On the other hand, as it is shown in Fig2(e), the authors achieved the switching even in 3.2ns. It might be difficult to reliably switch with such a short pulse time, but still 20 or 50ns should have been available. Anyway, I could not understand why the authors applied much longer 100 or 300ns pulses for the endurance test. If the authors cannot obtain the reliable endurance results using shorter pulse width, the device performance using Er-Sb₂Te₃ is not appealing at all.

4. Do the vertical axes “intensity” in Fig3(e) and (f) represent the contrast of the TEM image shown in (a)? Since Er is the heaviest element among three, it shows the brightest contrast. And also, Fig3(a) should not be a BF image but a dark-field (DF) image, because the van der Waals gaps look darker contrast.

5. Even though there are numerous calculation results in Fig4 and 5, as I mentioned in the comment 1, since the effects of f-electron were not taken into account, I wonder whether the results are reliable or not. The authors should compare the results with and without considering the effects of f-electron using LDA+U or any other techniques.

In summary, the most important device performances are not satisfactory, and most of the simulation results have concerns due to insufficient calculation parameters. These results brought me not to suggest the current version of manuscript for the publication in Nature Communications.

Manuscript: NCOMMS-20-47951-T

Title: “Designing Crystallization to Tune the Performance of Phase-Change Memory: Rules of Hierarchical Melt and Coordinate Bond”

Dear Reviewers,

Thank you for the reviewer’s comments concerning our manuscript. Those comments are all valuable and are very helpful for revising and improving our paper. We have studied the comments carefully and have made corrections which we hope will meet with approval. Revised portions are marked by red in the revised manuscript.

The response to the reviewer’s comments are as follows:

Referee 1:

J. Zhao et al. proposed a hierarchical melt and coordinate bond strategies to study the kinetic pathway of alloy-tuned crystallization in phase change material. They experimentally demonstrated Er-Sb₂Te₃ based phase change memories with 3.2 ns writing speed, ten-year data retention at 161 °C, 10⁵ endurance, 1.29 pJ power consumption and 0.41% density change rate. Some of the performances are among the best results in NVM field, like 3.2 ns operating speed and 1.29 pJ working energy. But some performances show obvious shortcoming, like 10⁵ endurance. The authors also conducted ab-initio calculation and discussed the crystallization kinetic of Er-Sb₂Te₃ material in-depth. In general, each part of this paper looks good, but the link between sections is weak. Here, I list one major issue and several technique issues for your reference:

Major issue:

In this work, the authors raised several critical issues in the field of phase change memory, including conflicted performances between writing speed and data retention, the detailed kinetic pathway of alloy-tuned crystallization, the gap between ab-initio simulation and experiment result, dopant selection criteria to improve data retention and speed etc. And they also provide plenty of experimental and simulation results to detailly discuss Er-Sb₂Te₃ material from multiple perspectives. However, the coupling between the critical issues, experimental results and ab-initio simulations is weak. In other words, the questions raised by authors are not fully addressed by the experimental and simulation results. For example:

(1) In experiment, the authors demonstrate 3.2 ns writing speed, but in simulation, the material is fully crystallized in 900 ps. The gap between experiment and ab-initio simulation still exists.

Thanks for your deep insights and valuable suggestions. Indeed, the speed predicted by ab-initio molecular dynamics simulation is faster than the experiment, which originates from the condition limits of the simulation, such as size effect and sample melting time. In this work, we focus on the inherent properties of materials, such as **the relative speed of different materials**, rather than the speed gap between simulation and experiment for one certain material. Using the simulation method, the crystallization time of $\text{Ge}_2\text{Sb}_2\text{Te}_5$ is 300ps~500ps. [1, 2] Using the same simulation conditions as $\text{Ge}_2\text{Sb}_2\text{Te}_5$, researchers cannot obtain the crystallization trajectories at the time scale of several hundred picoseconds, as no pre-existing nucleus is provided in the Sc-Sb₂Te₃ [3] and Y-Sb₂Te₃ [4] simulation systems. For our studied Er-Sb₂Te₃ system, the simulation crystallization time is ~900 ps. Thus, from the simulation results, the speed of Sc-Sb₂Te₃ and Y-Sb₂Te₃ as well as Er-Sb₂Te₃ systems should be slower than $\text{Ge}_2\text{Sb}_2\text{Te}_5$. However, all of them have faster chip speed than $\text{Ge}_2\text{Sb}_2\text{Te}_5$ with the experiment speed of ~30 ns [5], in which Sc-Sb₂Te₃ offers a record-breaking speed of 0.7 ns [3], and Er-Sb₂Te₃ has an impressive speed of 3.2 ns, and the operation speed of Y-Sb₂Te₃ achieves as fast as 6 ns [6]. A question or gap raised is that the relative speed of several different systems predicted by simulation is not in accordance with the relative experiment values.

These mistaken simulation results give us a suggestion that the crystallization processes of Er-, Sc-, and Y-doped Sb₂Te₃ are not the conventional annealing processes using the complete melt models. Here, using the “hierarchical melt” concept, the crystallized time of Er-Sb₂Te₃ is shortened to ~30 ps, assisted by the stabilized medium-range precursor. It is because empty Er 5*d* orbitals are shared with the lone-pair electrons of Te atoms near vacancies, which stabilizes a medium-range region around Er atoms near vacancies. The “hierarchical melt” concept is also suited to explain the faster speed of other systems, such as in Sc-Sb₂Te₃ and Y-Sb₂Te₃, because these dopants have empty *d* orbitals and stabilize the surrounding Te atoms by sharing the lone-pair electrons. The “hierarchical melt” concept thus bridges the gap of wrong relative speed predicted or tested by the simulation and experiment.

We are sorry for using a confusing word “gap”. We have revised the manuscript to emphasize the gap of relative-speed prediction, rather than the speed gap between simulation and experiment. We make the following changes in the Background Section, “The relative slower speed of YST than GST predicted by simulation is thus not in accordance with the relative experimental values. In other doped systems, their impressive faster chip speeds than GST are also hard to recur their crystallization trajectories by simulations.^{6, 9, 10, 11} Thus, a gap of the inconsistent results to predict the relative speed from the simulation and experiment should be bridged.”; and the changes in the last paragraph of the Results Section, “In the following, we utilize the hierarchical melt model to correct this mistake of relative speed prediction between the simulation and experiment ... Therefore, the hierarchical melt model provides a right simulation prediction that EST has faster speed than GST.”.

1. Lee TH, Elliott SR. Ab Initio Computer Simulation of the Early Stages of Crystallization: Application to $\text{Ge}_2\text{Sb}_2\text{Te}_5$ Phase-Change Materials. Phys Rev Lett 107, (2011).

2. Lee TH, Loke D, Elliott SR. Microscopic Mechanism of Doping-Induced Kinetically Constrained Crystallization in Phase-Change Materials. *Adv Mater* 27, 5477-5483 (2015).
3. Rao F, et al. Reducing the Stochasticity of Crystal Nucleation to Enable Subnanosecond Memory Writing. *Science* 358, 1423-1427 (2017).
4. Li Z, Si C, Zhou J, Xu H, Sun Z, Yttrium-Doped Sb₂Te₃: A Promising Material for Phase-Change Memory, *ACS Appl Mater Interfaces* 8, 26126-26134 (2016).
5. Wang Y, et al. Scandium doped Ge₂Sb₂Te₅ for high-speed and low-power-consumption phase change memory. *Appl Phys Lett* 112, 133104 (2018).
6. Liu B, et al. Y-Doped Sb₂Te₃ Phase-Change Materials: Toward a Universal Memory. *ACS Appl Mater Interfaces* 12, 20672-20679 (2020).

(2) The in-situ HRTEM shows very good atomic level resolution images of Er-Sb₂Te₃, but it only provides limited information that “Er locates at cationic positions”. It fails to provide a “direct atomic-level evidence” for “the detailed kinetic pathway of alloy-tuned crystallization”.

Thanks for this comment. While the HRTEM result manifests that Er atoms prefer the cationic positions in the crystalline phase, it is hard to monitor the local structures around the impurity atoms in the amorphous using general experimental techniques. Here, we use the atom-resolved simulation method to provide a “direct atomic-level evidence” for “the detailed kinetic pathway of alloy-tuned crystallization”. In Fig. 4(e)-(f), the calculation also provides the evidence that Er prefers the position near vacancy and the system presents lower energy, because lone-pair electrons nearby vacancy shown in Fig. 4(f) are shared with the empty Er *5d* orbitals. It is manifested by the more filled Er *5d* orbitals by the lone-pair electrons shown in Fig. 4g. Based on the calculation result that Er prefers positions near vacancies, we constructed the crystalline models, as shown in Fig. S13, which is used in the subsequent crystallization simulations. Subsequently, we do not completely melt the crystalline model constructed previously to obtain the amorphous model. To our surprise, the local region around two Er atoms as well as its nearby Te atoms is stabilized as the other region has been melt, namely the occurrence of hierarchical melt, as shown in Fig. 5g. In the following crystallization annealing process at 600 K, the stabilized medium-range region acts as a precursor to accelerate the crystallization, as shown in Fig. 5g-i, whose crystallization time achieves as fast as 30 ps. As a contrast, we monitor the crystallization trajectory of the complete melt model, in which the model melts fully at 3000 K for 120 ps and then quenches to 600 K for crystallization annealing. In the complete model, a series of cooperative movement is monitored during the crystallization and the finish time is as long as 900 ps, as shown in Fig. 5b-f. The hierarchical melt model successfully predicts that EST has faster speed than GST in the simulation, which is in line with the relative experiment values. Therefore, the hierarchical melt model provides a “direct atomic-level evidence” for “the detailed kinetic pathway of alloy-tuned crystallization”, namely Er atoms stabilizing a medium-range region to act as a precursor.

(3) The hierarchical melt and coordinate bond strategies is a good concept. But this concept should commonly exist in transition metal doped Sb_2Te_3 phase change materials, like $TiSbTe$, $ScSbTe$, $YSbTe$ etc.

Thank you for this valuable suggestion. The hierarchical melt and coordinate bond concepts are general to understand how replacement dopants influence the amorphous stability and crystallization processes. The calculation evidence in the main-text Fig. 4 has shown that the bonds of chalcogenide atoms become stronger by sharing its lone-pair electrons with the empty Er $5d$ orbitals. It is also suited to other dopants with empty orbitals, such as transition metals with empty d orbitals, to stabilize amorphous PCMs. [1, 2, 3] It is the same scenario in Al [4], Ga [5], In [6], and Sn [7] with more empty p orbitals to stabilize the chalcogenide glass, albeit their lower cohesive energy or low melting point compared with the transition metals.

In addition, similar to Er, a little mismatch metals, such as Sc [1], Y [2], and Ti [3] that exhibit the impressive transition speeds that have been reported, will prefer the cationic location near vacancies and present hierarchical melt as a little heat is provided. To verify this prediction, we utilize the model used in Fig. 4e to calculate the diffusion barrier of Sc- Sb_2Te_3 , Y- Sb_2Te_3 , and Ti- Sb_2Te_3 systems, as shown in Fig. R1. We obtain the similar results to EST that the much higher energy of FS (final state) structures. It illustrates that Sc, Y, and Ti atoms prefer the IS (initial state) positions that are near vacancies, because the vacancy-preferred lone-pair electrons, as shown in Fig. 4f, can be shared by the empty d orbitals as dopant atoms in the IS states, but less lone-pair electrons shared in FS states, as shown in Fig. R1d-f. It results in the relative higher

Figure R1 (a-c), the energetic profile of diffusion barrier for Sc, Y, Ti (black) or Sb (red and dash), where a model with a four-vacancy-aggregated cluster in the (111) plane is shown in the inner graph of Fig. 4e. (d-f), the partial DOS of Sc, Y, Ti d (black) and Te p (green) orbitals in the IS structure, while the Sc, Y, Ti d (red) orbitals in the FS structure also shown for comparison. It is noted that the pDOS of Te p orbitals is divided by Te number (48 herein).

energy of 0.71 eV for Sc, 0.79 eV for Y, and 0.62 eV for Ti than the reference model which just replaces the FS dopants by a Sb atom, as shown in Fig. R1a-c. These values are close to 0.75 eV in Er-Sb₂Te₃. Further, the medium-range crystal-like regions stabilized by these dopants will act as precursors to accelerate the crystallization.

We have made the above change in the first and second paragraphs of the main-text Discussion Part and Supplementary Fig. S14.

1. Rao F, et al. Reducing the Stochasticity of Crystal Nucleation to Enable Subnanosecond Memory Writing. *Science* 358, 1423-1427 (2017).
2. Liu B, et al. Y-Doped Sb₂Te₃ Phase-Change Materials: Toward a Universal Memory. *ACS Appl Mater Interfaces* 12, 20672-20679 (2020).
3. Zhu M, et al. One Order of Magnitude Faster Phase Change at Reduced Power in Ti-Sb-Te. *Nat Commun* 5, 4086-4086 (2014).
4. Peng C, et al. Al_{1.3}Sb₃Te Material for Phase Change Memory Application. *Appl Phys Lett* 99, 043105 (2011).
5. Kao K-F, Lee C-M, Chen M-J, Tsai M-J, Chin T-S. Ga₂Te₃Sb₅—A Candidate for Fast and Ultralong Retention Phase-Change Memory. *Adv Mater* 21, 1695-1699 (2009).
6. Saxena N, Persch C, Wuttig M, Manivannan A. Exploring Ultrafast Threshold Switching in In₃SbTe₂ Phase Change Memory devices. *Sci Rep* 9, 19251 (2019).
7. Bilovol V, et al. Structural, Vibrational and Electronic Properties in the Glass-Crystal Transition of Thin Films Sb₇₀Te₃₀ Doped with Sn. *J Alloys Compd* 845, 156307 (2020).

Technique issues:

1. The authors claimed that their ErSbTe-based phase change memory can serve as storage-class memory (SCM). As indicated by IBM research, there are 15 criteria to identify a SCM. The sub-10 ns speed, pJ level power consumption and 10-year@161oC thermal stability well meet the corresponding requirements of SCM. However, the write endurance of ErSbTe-based memory is only 10⁵, which can hardly meet the lifetime requirement of SCM (10⁹ – 10¹² write/erase cycles). Therefore, I think the authors should identify a new application scenario for this device.

Thanks for this useful comment. The previous 10⁵ endurance is not enough for the SCM application, but this performance strongly depends on the technological details, such as device cell, fabricating and testing processes. In the following, we explain how these factors influence the endurance in detail and take effective programs to improve them. (1) Effects of device cell. The PCRAM cell we used is a T-shaped structure with tungsten plug bottom electrode contact (BEC, a diameter of 190 nm shown in Fig. S4), which is fabricated using the 0.13- μ m-node CMOS technology. The BEC has a significant impact on device performance, because the cycle number of electrode itself determines the upper endurance limit of the PCRAM device. We test the previous cells without depositing phase-change materials, and the endurance is $\sim 4 \times 10^5$. Moreover, the electrode surface should be flatted using the chemical mechanical planarization (CMP). The accidented surface may result in the formation of cavity between the electrode and deposited film, due to the capillary effect, which can seriously reduce the endurance.

(2) Effect of fabrication process. In the etching process, each material needs appropriate parameters, which are always different for distinguished materials. Particularly, Er-Sb₂Te₃ film is difficult to be etched using the common parameters. Unreasonable etching parameters may damage the devices, and result in bad electrical test results.

(3) Effects of test equipment. For conducting the PCRAM cell, we should make the probe contact the top electrode, which produces different stress conditions. This process may damage and contaminate top electrode to result in the contact problems between the probe and top electrode, such as open circuit or electrical signal loss. In addition, the bad anti-vibration condition of the test bench reduces the probe/electrode contact stability, and results in low endurance.

Subsequently, we take effective programs to improve the above-mentioned potential disadvantages.

(1) We use a new T-shaped PCRAM cell, which is also fabricated using the 0.13- μ m-node CMOS semiconductor technology for the 190-nm-diameter tungsten plug BEC. The endurance of new devices is $\sim 1.5 \times 10^7$ without depositing phase-change material between two electrodes. We optimize the CMP technology to flat the electrode surface.

(2) We optimize the etching parameters (Ar, CF₄) to reduce the device damage.

(3) Burnish and clean the test probes in the test equipment. These can reduce signal loss during the testing process. In the end, we improve the anti-vibration condition.

Finally, we obtain the improved endurance about 10^7 (SET@50ns/2.2V and RESET@50ns/4.0V) for Er-Sb₂Te₃, as shown in Fig. R2. These demonstrate that device cell, fabricating and testing processes have important effects on the endurance performance. If the detailed technological processes can be improved further, such as using smaller BEC or confined structure, the endurance of Er-Sb₂Te₃ may increase further to meet the requirement of SCM ($10^9 - 10^{12}$ write/erase cycles).

We have made the corresponding changes in the Fig. 2e of the main text.

Figure R2 Operation cycles of Er_{0.52}Sb₂Te₃ under the conditions of SET @ 50ns/2.2V and RESET @ 50ns/4.0V.

2. For the power consumption calculation, the authors used a steady-state calculation method by using $Q_t = (V_{reset}^2/R_{set}) \times t_{reset} + (V_{set}^2/R_{reset}) \times t_{set}$. This method is based on the constant resistances of ON and OFF state. However, this calculation is questionable, because the resistance of phase change memory dynamically changes during resistive switching. I highly recommend to use transient calculation method by capturing current waveform during voltage pulse duration, which is more convincing for operating energy evaluation. Or the authors can also use the calculation method by using a current pulse source as in their previous works (Science 366, 210–215 (2019). Nature Communications 5 (1), 1-6 (2014)).

Thank you so much for pointing out the rough measurement of power consumption. The dynamical resistance occurs during the resistive switching of phase change memory, so the steady-state calculation using the formula of $Q_t = (V_{reset}^2/R_{set}) \times t_{reset} + (V_{set}^2/R_{reset}) \times t_{set}$ is not correct. Following your suggestion, we use the transient current pulse method [1, 2] to obtain the power consumption of Er-Sb₂Te₃. Firstly, we use the iterative current (*R-I* curve) to achieve the RESET state and find out the current value (1.9 mA), as shown in the inset of Figure R3. Then, apply a current pulse (500 ns width and 1.9 mA) to obtain RESET state again; at the same time, the oscilloscope equipment is used to capture the voltage *U* value (1.2 V). The power consumption is calculated according to the formula, $Q_{RESET} = I \times U \times t$, where *I* and *U* are the RESET current (1.9 mA herein) and voltage (1.2 V herein) values, respectively; and *t* is pulse width (500 ns herein). The calculated power consumption of Er-Sb₂Te₃ is 1.14 nJ, which is lower than 9.28 nJ of GST [3], 1.68 nJ of ScST [1], and 3.12 nJ of TiST [2].

We have made the corresponding changes in the Fig. 2d of the main text.

Figure R3. Using current pulse, the tested power consumption of Er_{0.52}Sb₂Te₃ is compared with GST [3], TiST [2], and ScST [1].

1. Rao F, et al. Reducing the stochasticity of crystal nucleation to enable subnanosecond memory writing. Science 358, 1423-1427 (2017).

2. Zhu M, et al. One order of magnitude faster phase change at reduced power in Ti-Sb-Te. Nat commun 5, 4086-4086 (2014).
3. Wang Y, et al. Scandium doped Ge₂Sb₂Te₅ for high-speed and low-power-consumption phase change memory. Appl Phys Lett 112, 133104 (2018).

3. In Fig. 1b, the authors mentioned that Er-Te has least mismatch with Sb₂Te₃ based on volume change rate calculation. I think their discussions in this part are weak. Because (1) lattice matching between two materials does not only depend on volume change rate, but also relate to other factors, like lattice constant, bond length, bond angle etc. (2) In Fig. 1b, the authors only included some transition metal elements, they missed some elements like Nb etc. for lanthanide series metals, they only calculated Er. So, it is hard to say Er-Te has the “least mismatch” with Sb₂Te₃. The authors should provide solid calculation results to support their discussion. This is important, because Er is not commonly used in standard semiconductor process. It should provide obvious advantages to motivate us to introduce new elements in standard process.

Thank you for your valuable suggestion. We modify the screening method to consider more parameters, such as lattice constants and bond lengths. We calculate most transition metals including lanthanides, as shown in Fig. R4. The mismatch table for the calculated tellurides (X₂Te₃) is shown in Table S1. From the figure and table, Er has the least mismatch with Sb₂Te₃.

We have made the corresponding changes in the Fig. 1b of the main text and Table S1 of the supplementary.

Figure R4. The lattice mismatch between metal tellurides and Sb₂Te₃, where Er has the least mismatch.

Table R1 Mismatch table for many tellurides (X_2Te_3)

Atomic number	Element	Average atomic volume ratio (%)	Lattice constant ratio in a- axis (%)	Lattice constant ratio in c-axis (%)	Bond length ratio (%)
13	Al	0.86967	7.36732	1.33651	8.08485
14	Si	1.26232	9.87052	0.20685	9.85018
21	Sc	0.58167	4.59593	0.28074	5.96787
22	Ti	1.51968	11.0994	2.31344	9.91803
23	V	2.09897	17.64116	1.50226	12.22132
24	Cr	2.17619	21.48509	11.00027	13.71504
25	Mn	2.37205	21.46618	4.44849	14.73796
26	Fe	2.57368	15.04249	14.93957	15.64138
27	Co	2.60197	14.99591	15.62	15.78856
28	Ni	2.44559	12.198	17.86927	15.18524
29	Cu	2.2008	8.01324	20.83269	12.03186
30	Zn	1.67548	6.56413	14.24338	8.02827
39	Y	0.4064	1.19222	3.61067	6.62475
40	Zr	0.81558	5.68875	1.0998	6.1941
41	Nb	1.83322	17.92872	7.63806	8.00681
42	Mo	1.88099	21.39192	16.85792	10.30989
43	Tc	2.12516	21.67764	11.0503	11.65115
44	Ru	2.07403	13.96211	6.93769	12.08246
45	Rh	2.08117	11.51333	12.15409	12.31458
46	Pd	1.956	6.44167	19.27489	10.62272
47	Ag	1.52979	3.70003	16.91206	6.27407
57	La	1.39227	7.71461	4.18855	7.23716
58	Ce	0.74724	3.64884	3.51654	5.1611
59	Pr	1.14927	6.00722	4.32814	6.88619
60	Nd	0.98783	5.17705	3.7925	6.74137
61	Pm	0.92286	4.02172	5.21021	2.17974
62	Sm	0.79511	3.35113	4.78599	6.70719
63	Eu	0.42887	2.97409	0.37381	4.96222
64	Gd	0.46122	1.9313	2.9051	6.67046
65	Tb	0.47503	1.61861	3.73995	6.57781
66	Dy	0.33316	0.94271	3.04542	6.59383
67	Ho	0.27894	0.48773	3.17521	6.62997
68	Er	0.01822	0.755	1.76232	2.45138
69	Tm	0.19825	0.05212	3.08112	6.68853
71	Lu	0.01604	0.85047	1.4782	6.40657
72	Hf	0.78354	5.80383	1.0386	7.05003
73	Ta	1.63227	19.38867	15.98341	8.1879
74	W	1.94082	22.45454	18.1846	9.83957
75	Re	2.12855	23.29601	15.69941	10.99302
76	Os	1.94733	11.98884	8.61033	12.06583
77	Ir	2.05571	9.1704	16.1645	12.23356
78	Pt	1.91552	7.02695	17.55292	10.83462
79	Au	1.59007	3.63615	17.99594	7.28907
83	Bi	0.48292	2.5198	2.03675	2.36692

4. In Figure 5, it can be seen that the Er atoms tends to form octahedrons. And two Er centered octahedrons stabilized medium-range region, which accelerates the crystallization. But in other doped Sb_2Te_3 phase change materials. But metal centered octahedrons are commonly formed in other phase change materials, like $Ti-SbTe$, $Sc-SbTe$, $Y-SbTe$ etc. So, the authors should indicate the difference between $Er-SbTe$ and other transition metal doped Sb_2Te_3 phase change materials.

Thanks for this comment. It has been reported the importance about the octahedral motifs around some dopants in the references [1, 2, 3], which were observed in the complete-melt models and exhibit their own random orientations. However, they ignore the necessary cooperative movement to form a critical nucleus, which should use a long

time to adjust the different orientations to the same. The much stable isolate octahedral motif still impedes the local structure movement, and slows down the formation of critical nucleus. It is the essential reason why the wrong relative speed is predicted in the Er-, Sc-, Y-, and Ti-doped systems using the complete melt models, albeit stable octahedral motifs observed in these systems.

To manifest the necessary of cooperative movement process in the experiment, we take a statistic of SET speed using the different melting time during the crystallization, whose operation details are shown in following paragraph. Figure R5 presents the box chart of SET speed. It exhibits the result that the longer melting time, the slower SET speed. It is because the longer melting time makes the amorphous more disorder and the system needs more cooperative movement processes, i.e., more time, to complete critical nucleation. It gives us a hint that less disorder amorphous or pre-existing medium-range crystal-like cluster can shorten the crystallization time. Therefore, we propose the “hierarchical melt” concept to produce a medium-range crystal-like region, which explains the faster chip speed of Er-Sb₂Te₃ and is also suited to Ti- Sb₂Te₃, Sc-Sb₂Te₃, and Sc- Sb₂Te₃ systems.

Figure R5. Box chart of SET speed under different melting time.

We use the iterative RESET operation to obtain the different melting states. We employ the incremental voltage pulses (0.1V herein) at a constant width, such as 100 ns in Fig. R6a and 1000 ns in Fig. R6b. Until the final pulse (brown) imposed, the previous pulses (blue) cannot melt the cell. Thus, the last critical pulse with different width and voltage value provides the different melting states. Subsequently, we immediately apply the SET pulses with gradually increased pulse width to test the SET speed, as shown in Fig. S8c. The “SET Speed” is defined as a critical pulse width after the sudden resistance reduce since the crystallization.

We have made the changes in the Fig. 2f of the main text and Fig. S8 of the supplementary. In the third paragraph of the Discussion Section, we also discuss the different concepts between our proposed “Hierarchical Melt” and conventional “metal centered octahedron”.

Figure R6 Different pulse waveforms applied to the PCRAM devices. **(a-b)**, iterative RESET operations with different pulse width to obtain different amorphous states. **(a)** and **(b)** obtain the short and long melting time rates, respectively. **(c)**, SET operations. The iterative SET pulse waveforms with the constant voltage are used to measure the SET speed.

1. Zhang W, Mazzarello R, Wuttig M, Ma E. Designing Crystallization in Phase-Change Materials for Universal Memory and Neuro-Inspired Computing. *Nat Rev Mater* 4, 150-168 (2019).
2. Rao F, et al. Reducing the Stochasticity of Crystal Nucleation to Enable Subnanosecond Memory Writing. *Science* 358, 1423-1427 (2017).
3. Zhou Y, et al. Bonding Similarities and Differences between Y-Sb-Te and Sc-Sb-Te Phase-Change Memory Materials. *J Mater Chem C* 8, 3646-3654 (2020).

Therefore, in my opinion, the authors should enhance and rearrange their experimental results and simulation results to make them better support each other.

Thank you for this helpful suggestion. We have revised the manuscript to emphasize the relative-speed-prediction gap between the simulation and experiment for different materials, which is solved by the “hierarchical melt” concept. We add the Discussion Part to highlight the importance and universality of the “hierarchical melt” concept.

Different from the conventional emphasized concept of metal-centered octahedron, we emphasize the importance of cooperative movement. It is the essential reason why the wrong relative speed is predicted using the complete melt models, albeit stable octahedral motifs observed in the system. Here, we provide the experimental evidence that the cooperative movement is necessary for the crystallization, as shown in Fig. R5. We take a statistic of SET speed using the different melting time. We find that the longer melting time, the slower SET speed. The processes of cooperative movement are also monitored in the main-text crystallization simulation, as shown in Fig. 5b-f.

Referee 2:

The manuscript by Zhao et al reports an alloy design of phase-change materials based on the idea on the hierarchical melt and coordinate bond. Even though there are some interesting results, the current version of manuscript is not suitable for publication due to the following reasons.

1. The authors mentioned that Er has the empty d orbitals. Does this mean 5d orbitals? On the other hand, Er is a lanthanide with partially filled f orbitals. In general, DFT calculations for f-electron materials are not straightforward. According to the Methods, the authors used just a conventional GGA function, so I wonder the reliability of their simulation results without using the DFT+U term.

5. Even though there are numerous calculation results in Fig4 and 5, as I mentioned in the comment 1, since the effects of f-electron were not taken into account, I wonder whether the results are reliable or not. The authors should compare the results with and without considering the effects of f-electron using LDA+U or any other techniques.

Thanks for your deep insights and helpful suggestions. Yes, Er has empty 5d orbitals, whose valence electrons are $Er_{3d} 5p^6 5d^1 6s^2$ used in the main text and the $4f^{11}$ electrons are treated as core states. Indeed, density functionals cannot handle f electrons well, due to self-interaction errors. A routine way to describe the localized 4f electrons by placing them in the core. In order to investigate the effect of Er 4f orbitals on the calculation results shown the main text, we consider Er 4f electrons as valence electrons to calculate the diffusion barrier of Er migration again, where the same model is used in Fig. 4. Figure R7 shows the relative energy and partial DOSs of the IS (initial state) and FS (final state) structures. As we use the Hubbard U value 4 eV to correct Er 4f electrons localization, the relative energy difference of 0.82 eV between FS1 and FS2 structures is similar to the value 0.75 eV shown in Fig. 4e. Without Hubbard U value added, the relative energy difference is 0.44 eV, which is about half value of 0.75 eV in Fig. 4e. Although the two scenarios give different relative values, they still predict that the Er atoms in FS structures are unstable. It is because the less vacancy-preferred lone-pair electrons fill the empty Er 5d orbitals, as shown in Fig. R7b and d. Finally, we calculate the partial DOS of Er s , p , d , and f orbitals, as shown in Fig. R7e and f. Although the energy of the 4f orbitals is close to the Fermi level, it is in fact the case that f-electrons do not play a major role in the bonding of rare-earth compounds. [1, 2] Many other calculations also obtained reasonable calculation results as Er 4f electrons in the core. [1, 3, 4, 5] It demonstrates that the pseudopotential performs well as the Er 4f electrons included in the core.

We have made the changes in the Discussion Section and Fig. S15.

Figure R7 **(a)-(b)**, the relative energy **(a)** and partial DOS **(b)** of the IS and FS1 structures, when the Hubbard U value of 4 eV is considered for Er f orbitals. **(c)-(d)**, the relative energy **(c)** and partial DOS **(d)** of the IS and FS structures, without Hubbard U value considered for Er f orbitals. **(e)-(f)**, the partial DOS of Er s , p , d , and f orbitals with Hubbard U value **(f)** or not **(e)**. The valence electrons in this figure are Er $5p^65d^16s^2$, Sb $5s^25p^3$, Te $5s^25p^4$.

1. Sanna, S., Hourahine, B., Gerstmann, U., Frauenheim, T. Efficient tight-binding approach for the study of strongly correlated systems. *Physical Review B* 76, 155128 (2007)
2. O'Donnell, K., Dierolf, V., Rare Earth Doped III-Nitrides for Optoelectronic and Spintronic Applications. 2010; Vol. 124.
3. Xu C, et al. A two-dimensional ErCu₂ intermetallic compound on Cu(111) with moire-pattern-modulated electronic structures. *Phys Chem Chem Phys* 22, 1693-1700 (2020).
4. Allan G, Lefebvre I, Christensen NE. Electronic structure of erbium disilicide. *Physical Review B* 48, 8572-8577 (1993).
5. van Setten MJ, et al. The PseudoDojo: Training and grading a 85 element optimized norm-conserving pseudopotential table. *Computer Physics Communications* 226, 39-54 (2018).

2. Device performance of four different Er-Sb₂Te₃ compositions were compared in Fig.2(e) and Fig. S2. I would recommend adding the results for the pure-Sb₂Te₃ to show the effects of Er-doping.

Thank you for this suggestion. Our previous manuscript lacks the device performance of the parental phase Sb₂Te₃, which is bad for directly showing the effect of Er dopant on the parental material. Following your suggestion, we test the device performance of pure Sb₂Te₃, as shown in Fig. R8, whose fastest speed is 6 ns. The speed of Er-Sb₂Te₃

we designed is two times faster than the pure Sb_2Te_3 . It is noted that its data retention of $161\text{ }^\circ\text{C}$ is much higher than the pure- Sb_2Te_3 of lower than $80\text{ }^\circ\text{C}$, which has a few small crystallites in the deposited film. Thus, the designed $\text{Er-Sb}_2\text{Te}_3$ presents the impressive properties of speed and data retention. Our designed $\text{Er-Sb}_2\text{Te}_3$ breaks through the empirical knowledge on the incompatible relationship between speed and data retention.

We have added the device performance of pure Sb_2Te_3 , as shown in Fig. S3c.

Figure R8 The SET-RESET windows for Sb_2Te_3 cell devices.

3. In Fig.2(f), the endurance data is shown, and the authors mentioned that “the tested endurance over 10^5 cycles, showing great potential for the storage-class memory applications”. I doubt that 10^5 is sufficient to use as the storage class memory, but at least 10^7 cycles are required.

Moreover, the authors applied 100ns/3.8V Reset and 300ns/1.8V Set pulses to switch the phases for the endurance test. On the other hand, as it is shown in Fig2(e), the authors achieved the switching even in 3.2ns. It might be difficult to reliably switch with such a short pulse time, but still 20 or 50ns should have been available. Anyway, I could not understand why the authors applied much longer 100 or 300ns pulses for the endurance test. If the authors cannot obtain the reliable endurance results using shorter pulse width, the device performance using $\text{Er-Sb}_2\text{Te}_3$ is not appealing at all.

Thank you very much for these useful comments. The previous 10^5 endurance is not enough for the SCM application, but this performance strongly depends on the technological details, such as device cell, fabricating and testing processes. In the following, we explain how these factors influence the endurance in detail and take effective programs to improve the endurance.

(1) Effects of device cell. The PCRAM cell we used is a T-shaped structure with tungsten plug bottom electrode contact (BEC, a diameter of 190 nm shown in Fig. S4), which is fabricated using the 0.13- μm -node CMOS technology. The BEC has a

significant impact on device performance, because the cycle number of the electrode itself determines the upper endurance limit of the PCRAM device. We test the previous cells without depositing phase-change materials, and the endurance is $\sim 4 \times 10^5$. Moreover, the electrode surface should be flatted using the chemical mechanical planarization (CMP). The accidented surface may result in the formation of cavity between the electrode and deposited film, due to the capillary effect, which can seriously reduce the endurance.

(2) Effect of fabrication process. In the etching process, each material needs appropriate parameters, which are always different for distinguished materials. However, Er-Sb₂Te₃ film is difficult to be etched using the common parameters. Unreasonable etching parameters may damage the devices, and result in bad electrical test results.

(3) Effects of test equipment. For conducting the PCRAM cell, we should make the probe contact the top electrode, which exhibits different stress conditions. This process may damage and contaminate top electrode to result in the contact problems between the probe and top electrode, such as open circuit or electrical signal loss. In addition, the bad anti-vibration condition for the test bench reduces the probe/electrode contact stability, and results in the low endurance.

Subsequently, we take effective programs to improve the above-mentioned potential disadvantages.

(1) We use a new T-shaped PCRAM cell, which is also fabricated using the 0.13- μm -node CMOS semiconductor technology for the 190-nm tungsten plug BEC. The endurance of new devices is $\sim 1.5 \times 10^7$ without depositing phase-change material between two electrodes. We optimize the CMP technology to flat the electrode surface.

(2) We optimize the etching parameters (Ar, CF₄) to reduce the device damage.

(3) Burnish and clean the test probes in the test equipment. These can reduce signal loss during the testing process. Finally, we improve the anti-vibration condition.

Finally, we obtain the improved endurance about 10^7 (SET@50ns/2.2V and RESET@50ns/4.0V) for Er-Sb₂Te₃, as shown in Fig. R9. Moreover, the endurance using a shorter pulse (SET@25ns/2.5V and RESET@15ns/3.8V) achieves 2×10^5 cycles. These demonstrate that device cell, fabricating and testing processes have important effects on the endurance performance. If the detailed technological processes can be improved further, such as using smaller BEC or confined structure, the endurance of Er-Sb₂Te₃ may increase hugely to meet the requirement of SCM ($10^9 - 10^{12}$ write/erase cycles).

We have made the changes in the Fig. 2e of the main text and Fig. S7 of the supplementary.

Figure R9 Operation cycles of $\text{Er}_{0.52}\text{Sb}_2\text{Te}_3$ under the conditions of (a) SET @ 25ns/2.5V and RESET @ 15ns/3.8V; (b) SET @ 50ns/2.2V and RESET @ 50ns/4.0V.

4. Do the vertical axes “intensity” in Fig3(e) and (f) represent the contrast of the TEM image shown in (a)? Since Er is the heaviest element among three, it shows the brightest contrast. And also, Fig3(a) should not be a BF image but a dark-field (DF) image, because the van der Waals gaps look darker contrast.

Thank you for your carefulness. We are sorry for this mistake in the manuscript. The bright-field (BF) image can be captured by the unscattered transmitted electron beam, as the diffracted electron beam is blocked by objective diaphragm. In contrast, when the diffracted electron beam passes through the objective diaphragm and the unscattered transmitted electron beam is blocked, the captured image is called the dark-field (DF) image. As you said that the heavier elements could show brighter contrast in the DF image, and the BF image shows the opposite contrast due to the difference in the electrons scattering of different atoms. In Fig. 3(a), Er is the heaviest element among three, and it shows the brightest contrast. So, it is indeed a DF image rather than a BF image. Thus, the vertical axes “intensity” in Fig. 3e-f represents the contrast of the TEM image shown in Fig. 3(a).

REVIEWERS' COMMENTS

Reviewer #2 (Remarks to the Author):

I think that the authors carefully addressed all my questions and comments, so now I believe that the manuscript should be published as it is.